# OpenReview forum: "Have the VLMs Lost Confidence? A Study of Sycophancy in VLMs"
_ICLR.cc/2025/Conference — ICLR 2025 Poster_

### Official Review · Reviewer_MDMB · 2024-11-01

**Soundness:** 3
**Presentation:** 4
**Contribution:** 3
**Rating:** 6
**Confidence:** 4

**Summary:**

This paper focuses on the study of sycophancy, a prevalent hallucination issue in Vision-Language Models (VLM). Firstly, a benchmark named MM-SY is introduced to evaluate the severity of sycophancy in VLMs. Subsequently, three methods—prompt guidance, Supervised Fine-Tuning (SFT), and Direct Policy Optimization (DPO)—are explored to mitigate sycophancy. Finally, the author analyzes the impact of attention weights on the sycophancy problem through experiments and proposes a simple, training-free method to alleviate this issue.

**Strengths:**

1. This paper is well written, clearly articulating the progressively detailed research approach to the sycophancy issue in VLMs.

2. Experiments are comprehensive, thoroughly testing multiple VLM models, various tasks, and different user preferences, and analyzing the relationship between sycophancy and various dimensions.

3. By studying the attention weights at different layers, this work reveals the model's performance in mitigating the sycophancy problem.

**Weaknesses:**

1. This paper mentions the contradiction between sycophancy and stubbornness issues, so for the VLM model, the real problem that needs to be addressed is to reduce sycophancy while maintaining the acceptance of correct opinions. However, methods such as prompt guidance, DPO, and amplify attention seem to reduce sycophancy but at the same time increase stubbornness to an equal extent. This does not truly solve the problem. It is merely shifting the imbalance from one side of the seesaw to the other. Only the SFT method shows a lower increase in stubbornness compared to the alleviation of flattery, but the paper does not provide a thorough analysis of this point.

2. This paper identifies the impact of amplifying high layer attention on the sycophancy problem but does not propose effective solutions based on this finding to truly address both sycophancy and stubbornness issues.

**Questions:**

As in Weaknesses, why does SFT perform better than other methods? Does high layer attention help in truly addressing the issues of flattery and stubbornness?

---

> ### Author Response · Authors · 2024-11-23
> **Official Response from Authors [1/2]**
>
> We are glad that the reviewer finds our paper well-written and our experimental results comprehensive. We address the reviewer's questions below.
>
> > **Response to W1 & Q1: This paper mentions the contradiction between sycophancy and stubbornness issues, so for the VLM model, the real problem that needs to be addressed is to reduce sycophancy while maintaining the acceptance of correct opinions. Why does SFT perform better than other methods?**
>
> - We greatly appreciate your insightful comment, which has encouraged us to conduct a deeper investigation and analysis of the stubbornness issue associated with sycophancy.
> - The key conclusion here is that **a model's correction ability stems from two aspects: its inherent helpfulness (beneficial) and its tendency to sycophancy (harmful)**. When correction ability decreases, we should delve deeper into whether this is caused by the mitigation of sycophancy or a decline in helpfulness (corresponding to the model becoming more stubborn). Following two recent works [1,2] that delve deeply into the sycophancy issue in pure-text LLMs, we have extended our correction experiments to identify the causes behind the decline in correction performance carefully.
> - We added a new experimental setup (hint without answer) to the original correction experiment (hint with answer). If a VLM’s correction ability stems from being helpful, it should be able to correct its answers under hints regardless of whether the answer is provided. In contrast, correction ability originating from sycophancy would struggle to work without an answer.
> - Results indicate that the correction ability of LLaVA-1.5-7B primarily derives from sycophancy (98.6 - 3.0 = 95.6), leaving almost no room for stubbornness in the model’s behavior. The SFT method not only mitigates sycophancy but also learns the correction ability from our constructed correction data (3.0 → 24.6). The DPO method, limited by the inherently low helpfulness of LLaVA-1.5-7B, achieves more thorough sycophancy mitigation but fails to enhance the model's correction ability through preference learning (3.0 → 0.1).
> - We also add experiments on InternVL-1.5-26B, which has a moderate level of inherent helpfulness (33.0). Under the SFT method, sycophancy is effectively mitigated, but helpfulness is also reduced (33.0 → 16.0). This could be due to the relatively lower quality of our constructed SFT data compared to InternVL’s original data in terms of task format and instruction diversity. The DPO method, however, not only mitigates sycophancy but also preserves and slightly enhances the model’s helpfulness (33.0 → 35.2).
> - In summary, for models like LLaVA-1.5-7B with very low inherent helpfulness, the SFT method mitigates sycophancy while improving helpfulness. For models like InternVL-1.5-26B with moderate helpfulness, the DPO method both mitigates sycophancy and enhances helpfulness. We will update the experimental results and provide a more comprehensive analysis of correction ability in the next version.
> | Model            | Syc↓   | Cor (hint w/ answer) | Cor (hint w/o answer) |
> | ---------------- | ---- | -------------------- | --------------------- |
> | LLaVA-1.5        | 94.6 | **98.6**                 | 3.0                   |
> | +SFT             | 25.4 | 42.1                 | **24.6**                  |
> | +DPO             | **5.4**  | 1.7                  | 0.1                   |
> | InternVL-1.5-26B | 90.6 | **98.6**                 | 33.0                  |
> | +SFT             | 18.2 | 19.2                 | 16.0                  |
> | +DPO             | **13.2** | 29.7                 | **35.2**                  |
>
> ---
>
> > [1]: Sharma, Mrinank, et al. "Towards Understanding Sycophancy in Language Models." The Twelfth International Conference on Learning Representations.
> >
> > [2]: Chen, Wei, et al. "From Yes-Men to Truth-Tellers: Addressing Sycophancy in Large Language Models with Pinpoint Tuning." Forty-first International Conference on Machine Learning.

---

> ### Author Response · Authors · 2024-11-23
> **Official Response from Authors [2/2]**
>
> > **Response to W2: This paper identifies the impact of amplifying high layer attention on the sycophancy problem but does not propose effective solutions based on this finding to truly address both sycophancy and stubbornness issues.**
>
> - Thank you for your question. We observe that enhancing high-level image attention in a training-free manner not only reduces sycophancy but also slightly improves the model's helpfulness (3.0 → 10.3, 11.2 → 12.7, 2.7 → 15.2). We would like to emphasize that this approach essentially serves as a test of the validity of our probing experiments and attention distribution analysis. While the sycophancy and correction performance is not state-of-the-art, it remains valuable given that it comes at nearly zero cost.
> | Model        | Syc  | Cor (hint w/ answer) | Cor (hint w/o answer) |
> | ------------ | ---- | -------------------- | --------------------- |
> | LLaVA-1.5    | 94.6 | **98.6**                 | 3.0                   |
> | +Amplified Image Attention L16-32    | **64.4** | 67.0                 | **10.3**                  |
> | BLIP-2       | 38.3 | **25.6**                 | 11.2                  |
> | +Amplified Image Attention L16-32    | **38.1** | 24.6                 | **12.7**                  |
> | InstructBLIP | 68.8 | **71.4**                 | 2.7                   |
> | +Amplified Image Attention L16-32    | **59.6** | 62.0                 | **15.2**                  |

---

> ### Author Response · Authors · 2024-12-03
> **Kind Reminder from Authors**
>
> Dear Reviewer MDMB,
>
> Thank you for your valuable suggestions. We would like to kindly remind you regarding the response to our rebuttal for the paper. We deeply value your insights and constructive feedback, as they are instrumental in improving the quality of our work.
>
> As the deadline for finalizing decisions approaches, we would greatly appreciate it if you could share any further comments or recommendations at your earliest convenience. We understand your time is valuable and are sincerely grateful for your dedication to the review process.
>
> Please let us know if there’s any additional information we can provide to assist you.
>
> Thank you once again for your time and effort.
>
> Warm regards,
>
> Authors

---

### Official Review · Reviewer_eeSv · 2024-11-04

**Soundness:** 2
**Presentation:** 2
**Contribution:** 2
**Rating:** 6
**Confidence:** 4

**Summary:**

The paper "Have the Vision-Language Models Lost Confidence? A Study of Sycophancy in VLMs" introduces the concept of sycophancy in vision-language models (VLMs), where models blindly agree with user inputs despite contradictory visual evidence. The authors present the MM-SY benchmark, the first evaluation benchmark for sycophancy in VLMs across ten visual understanding tasks. They find that VLMs exhibit significant sycophantic behavior, influenced by factors like task type, user tone, and model size. To address this, the paper explores three mitigation methods: prompt-based, supervised fine-tuning, and direct preference optimization, showing progressive improvements in reducing sycophancy. However, these methods also make VLMs more resistant to corrections. The authors propose a training-free approach by amplifying high-layer vision attention, which effectively mitigates sycophancy without compromising the model's receptiveness to corrections.

**Strengths:**

1. The paper offers a thorough analysis of the factors influencing sycophancy in VLMs, providing valuable insights into model behavior across different conditions.

2. The exploration of three distinct mitigation methods, each with varying degrees of success, contributes to the understanding of how to manage sycophantic behavior in VLMs.

3. The proposal of a simple, training-free method to reduce sycophancy by amplifying high-layer vision attention is innovative and has practical implications for model development.

**Weaknesses:**

1. The mitigation methods were only validated on a single VLM (LLaVA-1.5-7B), which limits the generalizability of the findings. It's unclear how these methods would perform across different VLM architectures.

2. The paper mentions that due to time and computational resource constraints, the analysis was limited. This suggests that the findings may not be exhaustive and could benefit from further exploration with additional resources.

**Questions:**

Please refer to weaknesses.

---

> ### Author Response · Authors · 2024-11-23
> **Official Response from Authors**
>
> We are glad the reviewer finds our proposed benchmark to be novel and our analysis of the factors influencing sycophancy to be thorough. We respond to the reviewer's questions below.
>
> > **Response to W1: It's unclear how these methods would perform across different VLM architectures.**
>
> - Thanks for the great comment, we agree that this was missing in the original version of the manuscript. We are excited to have added new results on InternVL-1.5-26B, demonstrating the consistent effectiveness of our method.
>   | Model            | Acc@R1 | Syc$\downarrow$ | Cor (hint w/ answer)  |
>   | ---------------- | ------ | --------------- | ---- |
>   | InternVL-1.5-26B | 93.2   | 90.6            | **98.6** |
>   | +Prompt          | 93.1   | 77.7            | 94.7 |
>   | +SFT             | 92.1   | 18.2            | 19.2 |
>   | +DPO             | **93.7**   | **13.2**            | 29.7 |
>
> - We observe that, similar to LLaVA-1.5-7B, the original InternVL-1.5-26B model exhibit significant sycophancy (90.6 Syc). Our three mitigation methods—Prompt, SFT, and DPO—were all effective in reducing sycophancy. The Prompt method mitigate sycophancy to some extent (-12.9 Syc). SFT effectively mitigate sycophancy by -72.4 Syc, though the correction rate remain relatively low (19.2 Cor). DPO demonstrated the most substantial mitigate in sycophancy (-77.4 Syc) and result in a higher correction rate (29.7 Cor vs 19.2 Cor), outperforming SFT.
>
> - These results highlight that our proposed methods generalize well across different VLM architectures, consistently improving sycophancy mitigation. We hope this additional experiment addresses the reviewer’s concerns about the generalizability of our approach.
>
> > **Response to W2: The paper mentions that due to time and computational resource constraints, the analysis was limited.**
>
> - Thanks for your comment. We conduct additional experiments to explore the relationship between sycophancy and model performance.
>
> - Firstly, we analyze 10 VLMs with diverse downstream task performances and sycophancy rates, ranking them by their average accuracy across comprehensive downstream benchmarks. Our findings reveal no clear relationship between sycophancy levels and baseline accuracy.
> | Model                    | Acc@1 | Syc$\downarrow$  |
> | ------------------------ | ----- | ---- |
> | BLIP2                    | 71.9  | 38.3 |
> | Gemini                   | 74.9  | 59.8 |
> | InstructBLIP             | 78.0  | 68.8 |
> | LLaVA-1.5                | 84.7  | 94.6 |
> | mPLUG-Owl2               | 86.8  | 66.0 |
> | GPT-4V                   | 89.3  | 39.4 |
> | InternLM-XC2-1.8B        | 90.7  | **28.8** |
> | InternVL-1.5-26B         | 93.2  | 90.6 |
> | InternVL-1.5-2B          | 93.3  | 80.2 |
> | InternLM-XC2-7B          | **94.0**  | 39.8 |
>
> - Secondly, using the same VLM (LLaVA-1.5), we find that while our SFT and DPO methods substantially mitigate the sycophancy rate, the model's performance on general tasks—including MM-SY downstream tasks and six general benchmarks remain unaffected. These results demonstrate that sycophancy mitigation can be achieved without compromising general task performance.
> | Model     | Syc$\downarrow$ | Acc@1 | SEED${^I}$ | POPE | SQA${^I}$ | MMBench | MMBench$^{CN}$ | MMVet | Avg@6 |
> | --------- | --------------- | ----- | ---------- | ---- | --------- | ------- | -------------- | ----- | ----- |
> | LLaVA     | 94.6            | 84.7  | **66.2**       | 85.9 | 66.8      | 63.0    | 57.4           | 30.5  | 61.6  |
> | +Amplified Image Attention L16-32 | 64.4            | **88.3**  | 64.8       | 83.8 | 65.8      | 64.4    | 57.6           | **31.7**  | 61.4  |
> | +SFT      | 25.4            | 88.1  | 65.2       | **86.6** | 67.5      | **66.1**    | **59.1**           | 29.6  | **62.3**  |
> | +DPO      | **5.4**             | 84.3  | 65.2       | **86.6** | **67.8**      | 65.9    | 59.0           | 28.9  | 62.2  |

---

> ### Author Response · Authors · 2024-12-03
> **Kind Reminder from Authors**
>
> Dear Reviewer eeSv,
>
> Thank you for your valuable suggestions. We would like to kindly remind you regarding the response to our rebuttal for the paper. We deeply value your insights and constructive feedback, as they are instrumental in improving the quality of our work.
>
> As the deadline for finalizing decisions approaches, we would greatly appreciate it if you could share any further comments or recommendations at your earliest convenience. We understand your time is valuable and are sincerely grateful for your dedication to the review process.
>
> Please let us know if there’s any additional information we can provide to assist you.
>
> Thank you once again for your time and effort.
>
> Warm regards,
>
> Authors

---

### Official Review · Reviewer_CV64 · 2024-11-04

**Soundness:** 4
**Presentation:** 3
**Contribution:** 3
**Rating:** 8
**Confidence:** 3

**Summary:**

This paper investigates the sycophancy problem in VLMs, which is also a common hallucination issue in LLMs. The authors first design an evaluation benchmark along with 10 visual question answering (VQA) tasks to assess the sycophancy problem in popular VLMs. They then propose three methods from the perspective of prompt engineering, supervised fine-tuning, and direct preference optimization to mitigate this issue.

**Strengths:**

+ This is the first paper to investigate the hallucination problem in multi-modality language models. To address this issue, the authors construct a new evaluation benchmark that includes 10 different visual question answering (VQA) tasks.

+ Based on the designed benchmark, the authors investigate this problem on various popular VLMs and provide comprehensive experimental results.

+ Besides revealing the sycophancy phenomenon, the authors also provide three different kinds of solution to alleviate this hallucination problem.

**Weaknesses:**

+ It seems that the definition of sycophancy rate is missing. Could the authors present it in Section 2? This is important for the readers to understand Table 1 and Figure 2.

+ In addition to revealing the sycophancy phenomenon, it would be beneficial to analyze why the current model tends to exhibit sycophancy. For example, is this comes from the training data or the network architecture?"

**Questions:**

+ From Table 1, it seems the sycophancy rate is not correlated to the designed types of tones. Could the authors provide the analysis for this?
+ In addition to the sycophancy rate, it would be beneficial to also present the baseline accuracy. It is interesting to see if the model's sycophancy rate related to its performance?

---

> ### Author Response · Authors · 2024-11-23
> **Official Response from Authors [1/2]**
>
> We are glad the reviewer finds our proposed benchmark to be novel and our experimental results to be comprehensive. We respond to the reviewer's questions below.
>
> > **Response to W1: The definition of sycophancy rate is missing**
>
> The sycophancy rate is calculated as:
>
> $\text{Sycophancy Rate} = \frac{\sum_{i=1}^N I(A_i^{(2)} == U_i^{(2, neg)})}{N}$,
>
>  where:
> - $A_i^{(2)}$ represents the second-round answer given by the VLM for the $i$-th sample.
> - $U_i^{(2, neg)}$ is the incorrect opinion provided by the user for the $i$-th sample.
> - $I(\cdot)$ is an indicator function that equals 1 if $A_i^{(2)}$ matches $U_i^{(2, neg)}$, and 0 otherwise.
> - $N$ is the total number of samples in the MM-SY benchmark.
>
> It quantifies the percentage of instances where the model conforms to the user's incorrect viewpoint (given that the first-round response was correct), thereby reflecting the extent of the model's sycophancy.
>
>
> > **Response to W2: It would be beneficial to analyze why the current model tends to exhibit sycophancy. E.g., is this comes from the training data or the network architecture?"**
>
> - Thanks for the great comment. Although the causes of sycophancy in VLMs remain unexplored, we attempt to conduct some preliminary discussions by drawing on the causes of sycophancy in text-only LLMs.
> - [1] suggests that sycophancy arises from human preferences during the RLHF process. However, LLaVA, which uses Vicuna-v1.5 (a model not trained with RLHF) as its initialization, still demonstrates a sycophancy rate as high as 94.6. Therefore, we argue that RLHF is not a necessary condition for sycophancy to occur.
> - We list the characteristics of 10 evaluated VLMs (e.g., image resolution, presence of SFT) and attempt to analyze the potential underlying reasons.
>   - We argue that image resolution is not a necessary condition for sycophancy. BLIP-2 and InstructBLIP have same image resolution, but the sycophancy rate of InstructBLIP is higher than that of BLIP-2. InternVL-1.5 has a higher image resolution than LLaVA-1.5, but they both have a sycophancy rate over 90.
>   - We suggest that original instruction tuning might be responsible for sycophancy. InstructBLIP uses BLIP-2 as its initialization and performs instruction tuning. Its sycophancy rate is much higher than that of BLIP-2. The model may confuse helping a user with a task with sycophancy. Adding the sycophancy suppression data proposed in this paper to the original instruction fine-tuning dataset may be one of the mitigation solutions. This will be part of our future work. Thank you again for your comments.
> | Model             | Syc$\downarrow$  | w/ RLHF-LLM | Resolution | w/ Instruction data |
> | ----------------- | ---- | ----------- | ---------- | ------------------- |
> | BLIP-2            | 38.3 | N           | 224        | N                   |
> | InstructionBLIP   | 68.8 | N           | 224        | Y                   |
> | LLaVA-1.5         | 94.6 | N           | 336        | Y                   |
> | mPLUG-Owl2        | 66.0 | N           | 224        | Y                   |
> | InternVL-1.5-2B   | 80.2 | N           | Dynamic    | Y                   |
> | InternVL-1.5-26B  | 90.6 | N           | Dynamic    | Y                   |
> | InternLM-XC2-1.8B | 28.8 | N           | Dynamic    | Y                   |
> | InternLM-XC2-7B   | 39.8 | N           | Dynamic    | Y                   |
> | Gemini            | 59.8 | Unknown     | Unknown    | Y                   |
> | GPT-4V            | 39.4 | Unknown     | Unknown    | Y                   |
>
> ---
>
> > [1]: Sharma, Mrinank, et al. "Towards Understanding Sycophancy in Language Models." The Twelfth International Conference on Learning Representations.

---

> ### Author Response · Authors · 2024-11-23
> **Official Response from Authors [2/2]**
>
> > **Response to Q1: It seems the sycophancy rate is not correlated to the designed types of tones**
>
> - Thank you for your comment. We design three tone types to avoid biased results caused by using a single template for evaluation. The conclusion is that there is no strong correlation between sycophancy and tone type. Even with a Euphemistic tone, sycophancy remains highly prevalent.
> - Lines 192–196: We also analyze the performance of specific VLMs under different tones and find that there is still no strong correlation.
>
> > **Response to Q2: It would be beneficial to also present the baseline accuracy. It is interesting to see if the model's sycophancy rate related to its performance?**
>
> - Thank you for your great comment. We present the relationship between sycophancy in VLMs and their performance from two perspectives.
> - Firstly, we examine different VLMs, which have varying downstream task performances and sycophancy rates. We rank 10 VLMs based on their average performance on comprehensive downstream tasks. No obvious correlation is observed between sycophancy and baseline accuracy.
> | Model                    | Acc@1 | Syc$\downarrow$  |
> | ------------------------ | ----- | ---- |
> | BLIP2                    | 71.9  | 38.3 |
> | Gemini                   | 74.9  | 59.8 |
> | InstructBLIP             | 78.0  | 68.8 |
> | LLaVA-1.5                | 84.7  | 94.6 |
> | mPLUG-Owl2               | 86.8  | 66.0 |
> | GPT-4V                   | 89.3  | 39.4 |
> | InternLM-XC2-1.8B        | 90.7  | **28.8** |
> | InternVL-1.5-26B         | 93.2  | 90.6 |
> | InternVL-1.5-2B          | 93.3  | 80.2 |
> | InternLM-XC2-7B          | **94.0**  | 39.8 |
>
> - Secondly, for the same VLM (LLaVA-1.5), although our SFT and DPO methods significantly mitigate the sycophancy rate, the VLM's performance on general tasks (whether on MM-SY downstream tasks or the six general benchmarks like MMBench) is not affected.
> | Model     | Syc$\downarrow$ | Acc@1 | SEED${^I}$ | POPE | SQA${^I}$ | MMBench | MMBench$^{CN}$ | MMVet | Avg@6 |
> | --------- | --------------- | ----- | ---------- | ---- | --------- | ------- | -------------- | ----- | ----- |
> | LLaVA     | 94.6            | 84.7  | **66.2**       | 85.9 | 66.8      | 63.0    | 57.4           | 30.5  | 61.6  |
> | +Amplified Image Attention L16-32 | 64.4            | **88.3**  | 64.8       | 83.8 | 65.8      | 64.4    | 57.6           | **31.7**  | 61.4  |
> | +SFT      | 25.4            | 88.1  | 65.2       | **86.6** | 67.5      | **66.1**    | **59.1**           | 29.6  | **62.3**  |
> | +DPO      | **5.4**             | 84.3  | 65.2       | **86.6** | **67.8**      | 65.9    | 59.0           | 28.9  | 62.2  |

---

> ### Author Response · Authors · 2024-12-03
> **Kind Reminder from Authors**
>
> Dear Reviewer CV64,
>
> Thank you for your valuable suggestions. We would like to kindly remind you regarding the response to our rebuttal for the paper. We deeply value your insights and constructive feedback, as they are instrumental in improving the quality of our work.
>
> As the deadline for finalizing decisions approaches, we would greatly appreciate it if you could share any further comments or recommendations at your earliest convenience. We understand your time is valuable and are sincerely grateful for your dedication to the review process.
>
> Please let us know if there’s any additional information we can provide to assist you.
>
> Thank you once again for your time and effort.
>
> Warm regards,
>
> Authors

---

### Author Response · Authors · 2024-11-26
**General Comment**

We would like to thank the reviewers for their efforts in assessing our work and the valuable feedback. We have accordingly made significant improvements following the suggestions and comments. For convenience, *we have marked in the revised PDF in blue* the added sections and the added results in tables. Light blue prefixes indicate the Reviewer ID and question ID for easy reference (it will not appear in the next version).

> R1 corresponds to Reviewer CV64.
>
> R2 corresponds to Reviewer eeSv.
>
> R3 corresponds to Reviewer MDMB.

Below we summarise our changes:
- **Formal definition of sycophancy:** Following Reviewer CV64's request, we have added this in Appendix A.2.
- **Discussion on potential causes of sycophancy:** Following Reviewer CV64's request, we have provided a discussion in Appendix A.5, covering aspects such as whether the LLM underwent RLHF training, image resolution, the use of image-text interleaved data, and multimodal SFT training.
- **Relationship between sycophancy and tone:** Following Reviewer CV64's request, we have emphasized this in Section 2.2 RQ2.
- **Relationship between sycophancy and baseline performance:** Following the requests from Reviewers CV64 and eeSv, we have added analyses in Appendix A.6, including:
  1. A comparison of sycophancy and baseline performance across multiple VLMs.
  2. An analysis of baseline performance for the same VLM (LLaVA-1.5) under different sycophancy mitigation methods.
- **Sycophancy mitigation experiments for InternVL-1.5-26B:** Following Reviewer eeSv's request, we have included InternVL-1.5-26B in the main results in Table 2.
- **New setup for correction experiments:** Following Reviewer MDMB's request, we have expanded the correction experiments to include a new setup where prompts do not contain answers (only indicating that the first-round answer is incorrect). This effectively differentiates the correction ability derived from *sycophancy* and the *helpfulness* of the VLM.
- **New correction experiments for the Amplified Image Attention method across three VLMs (Cor w/o Answer):** Following Reviewer MDMB's request, we have extended the correction experiments for the proposed Amplified Image Attention method. The results demonstrate that this method can mitigate sycophancy and slightly enhance correction ability—or, more accurately, the "helpful ability" of the VLM—in a training-free manner, validating our hypothesis that "high-level attention lacks focus on image tokens."

We hope that our revisions have strengthened our contributions and would like to thank the reviewers for their valuable suggestions. We look forward to productive rebuttal discussions.

---

### Meta-Review · Area_Chair_x6nH · 2024-12-15

**Metareview:**

This paper investigates the hallucination problem, specifically sycophancy, in multi-modality language models (VLMs) by constructing a new benchmark for evaluating 10 different VQA tasks. The paper provides comprehensive experiments across popular VLMs, offering valuable insights into sycophantic behavior and the factors influencing it. It proposes three mitigation methods, with varying success, and introduces a novel, training-free approach to reduce sycophancy by amplifying high-layer vision attention, demonstrating both theoretical and practical contributions.

All reviewers recognize the contributions of the paper, with feedback generally leaning towards acceptance. The AC thoroughly reviewed the paper and rebuttal, agreeing with the consensus recommendation for acceptance due to the consistency of the feedback.

**Additional Comments On Reviewer Discussion:**

The reviewers noted that most of the concerns raised have been addressed, leading to a unanimous recommendation for acceptance.

---

### Decision · Program_Chairs · 2025-01-22

Accept (Poster)